# Probiotic Potential *Lacticaseibacillus casei* and *Limosilactobacillus fermentum* Strains Isolated from Dosa Batter Inhibit α-Glucosidase and α-Amylase Enzymes

**DOI:** 10.3390/microorganisms10061195

**Published:** 2022-06-11

**Authors:** Chandana Kumari V. B., Sujay S. Huligere, Abdullah M. Shbeer, Mohammed Ageel, Jayanthi M. K., Jagadeep Chandra S., Ramith Ramu

**Affiliations:** 1Department of Biotechnology and Bioinformatics, School of Life Sciences, JSS Academy of Higher Education and Research, Mysuru 570015, Karnataka, India; chandanavb2@gmail.com (C.K.V.B.); sujayhuligere@gmail.com (S.S.H.); 2Department of Surgery, Faculty of Medicine, Jazan University, Jazan 45142, Saudi Arabia; ashbeer@jazanu.edu.sa (A.M.S.); ageelahmed@jazanu.edu.sa (M.A.); 3Department of Pharmacology, JSS Medical College, JSS Academy of Higher Education and Research, Mysuru 570015, Karnataka, India; mkjayanthi@jssuni.edu.in; 4Department of Microbiology, School of Life Sciences, JSS Academy of Higher Education and Research, Mysuru 570015, Karnataka, India; jagadeepchandramys@gmail.com

**Keywords:** α-glucosidase, α-amylase, probiotics, lactic acid bacteria, dosa batter

## Abstract

Fermented food plays a major role in gastrointestinal health, as well as possesses other health benefits, such as beneficiary effects in the management of diabetes. Probiotics are thought to be viable sources for enhancing the microbiome of the human gut. In the present study, using biochemical, physiological, and molecular approaches, the isolated *Lactobacillus* spp. from dosa batter were identified. The cell-free supernatant (CS), cell-free extract (CE), and intact cells (IC) were evaluated for their inhibitory potential against the carbohydrate hydrolyzing enzymes α-glucosidase and α-amylase. Then, 16S rDNA amplification and sequencing were used to identify the species. A homology search in NCBI database was performed that suggests the isolates are >95% similar to *Limosilactobacillus fermentum* and *Lacticaseibacillus casei*. Different standard parameters were used to evaluate the probiotic potential of strains RAMULAB07, RAMULAB08, RAMULAB09, RAMULAB10, RAMULAB11, and RAMULAB12. The strains expressed a significant tolerance to the gastric and intestinal juices with a higher survival rate (>98%). A high adhesion capability was observed by the isolates exhibited through hydrophobicity (>65%), aggregation assays (>75%), and adherence assay on HT-29 cells (>82%) and buccal epithelial cells. In addition, the isolates expressed antibacterial and antibiotic properties. Safety assessments (DNase and hemolytic assay) revealed that the isolates could be classified as safe. α-glucosidase and α-amylase inhibition of the isolates for CS, CE, and IC ranged from 7.50% to 65.01% and 20.21% to 56.91%, respectively. The results suggest that these species have exceptional antidiabetic potential, which may be explained by their use as foods that can have health-enhancing effects beyond basic nutrition.

## 1. Introduction

Diabetes mellitus (DM) is a long-term metabolic condition marked by persistent hyperglycemia. It could be due to a lack of insulin production, resistance to insulin, or both [1]. Small-intestinal α-glucosidase and pancreatic α-amylase are the key enzymes of carbohydrate digestion in humans. Antagonists of these enzymes may be helpful in preventing postprandial hyperglycemia by delaying carbohydrate digestion and glucose absorption [2,3]. In this regard, the isolated lactic acid bacteria (LAB) obtained from various sources are commonly present in the gut and play an important role in certain features of diabetes, as per recent research [4,5]. As a result, modifying the intestinal microbiota to attain, restore, and maintain a favorable ecological balance, as well as the activity of bacteria in the gastrointestinal tract, is crucial for the hosts’ enhanced health [6]. *Lactobacillus* spp. lower α-glucosidase in vitro by delaying carbohydrate metabolism (ingestion and absorption), resulting in hypoglycemia, improved glucose equilibrium, pancreatic function, insulin resistance, and alleviation of the associated oxidative damage [7]. The *Lactobacillus* spp. colonization in the gut is required, but they must also have an innate ability to stabilize acid-bile, resist digestive enzymes, increase food solubility, restore mucosal integrity, and produce of vitamins and enzymes [8]. These properties of LAB facilitate them for survival in the gastric and bile conditions to adhere to the host intestinal epithelium and are important for bacterial maintenance in the human gastrointestinal tract. This property could be important in pathogenic bacteria’s competitive exclusion. LAB is a major component of the human gastrointestinal tract’s mutualistic microbial flora and is commonly used as probiotics as well as in the fermentation of food products [9].

In South Indian cuisine, there are many traditional fermented foods customarily consumed in day-to-day life. Rice is India’s main cereal, and it is consumed in large quantities. There are many dishes made up of rice such as idli, dosa, appam, etc. [10]. Dosa is one such popular dish prepared from fermented batter including primarily black gram and rice [11]. Fermentation improves the product’s sensory qualities also in addition to the removal of undesirable constituents, making nutrients more accessible while preserving and increasing the levels of many bioactive compounds. Co-fermentation of cereals and legumes has been suggested as a way to make low-cost protein-rich foods while also improving the macro- and micronutrient balance of cereal-based fermented foods [12]. A majority of fermented foods and traditional Indian cuisines are fermented by LAB along with cereals and legumes [13]. *Lactococcus* spp. isolated from dosa batter is proven to have nisin-producing ability as well as expresses antibacterial activity against foodborne pathogens [14]. Previously, antibiotic susceptibility of *Pediococcus* spp. isolated from dosa batter was beneficial in their selection for future food and feed applications [15]. The chemical compound (2-hydroxyl indole-3-propanamide) derived from LAB strains isolated from idly and uttapam batter has potential antibiotic properties that protect probiotic bacteria against the pathogenic strains [16]. With this background, the study objectives are to isolate LAB from dosa batter that has probiotic potential and study its inhibitory potential against the carbohydrate hydrolyzing enzymes (α-glucosidase and α-amylase).

## 2. Materials and Methods

### 2.1. Materials

*Lactobacillus* de Man, Rogosa, and Sharpe (MRS) agar and broth, glycerol, NaCl, oxgall salt, xylene, phenol, blood agar medium with 5% (*w*/*v*) sheep blood, antibiotic susceptibility disc, ABTS, DPPH, and deoxyribonuclease (DNase) agar medium required for this study were obtained from HiMedia Laboratories Pvt. Ltd., Mumbai, India. The pathogens (*Escherichia coli* MTCC 4430, *Bacillus subtilis* MTCC 10403, *Micrococcus luteus* MTCC 1809, *Pseudomonas aeruginosa* MTCC 424, and *Salmonella typhimurium* MTCC 98) were obtained from Microbial Type Culture Collection and Gene Bank (MTCC), Chandigarh, India. Instruments used were microplate reader Multiskan FC, CO_2_ incubator (Thermo Fisher Scientific, Mumbai, India). The centrifugation unit used in this study was the REMI refrigerated micro centrifuge (RPM 7000, C-30 Plus; relative centrifugal force: 20,000× *g* to 22,000× *g*).

#### Preparation of Sample and Isolation of Lactic Acid Bacteria

Dosa batter sample was prepared by soaking rice and black gram (3:1) in water overnight and pulverized finely. This was allowed to ferment overnight and used to isolate lactic acid bacteria strains. The serially diluted batter 100 µL dilution was plated on MRS agar (de Man, Rogosa, and Sharpe agar) plates (37 °C, 24–48 h). By observation, idiosyncratic colonies were isolated and initially scrutinized using Gram’s staining, and catalase tested to select the culture (37 °C, 24 h) [16,17]. For the assays, the isolates were grown on MRS broth overnight (18 h) and were centrifuged at 2000 rpm for 10 min at 4 °C. Pellets were collected, washed twice with phosphate buffer saline (PBS) (pH 7.4), and the obtained cells were adjusted to 1 × 10^8^ CFU/mL. The calibration for each isolate was performed prior to adjusting the concentration to 0.5 McFarland standards turbidity by measuring its optical density at 625 nm [18].

### 2.2. Preliminary Assay of Lactic Acid Bacteria (LAB)

The preliminary biochemical assay on the isolated LAB was carried out according to the principles of Bergey’s manual of determinative bacteriology using different tolerable temperature, NaCl concentration (2%, 3%, 7%, and 10% NaCl), pH (2, 4, 6, and 7.4) tolerance, and carbohydrates fermentation. The viability and survival rate of the isolates in the presence of phenol solution were investigated by method described by Jena et al. (2013) [19] by inoculating the LAB isolated (108 CFU/mL) in MRS broth containing 0.4% phenol (24 h, 37 °C). At initial 0 h and after incubation for 24 h, the bacterial enumeration was performed by serial dilution on to the MRS agar plate.

### 2.3. Probiotic Properties

#### 2.3.1. Adherence Assay

##### Cell Surface Hydrophobicity

The cell surface hydrophobicity of the isolates was tested according to the method described by Botthoulath et al. (2018) [20] with slight modification. It helps us understand the interaction between xylene (polar solvent) and the bacteria. A total of 3 mL of 10^8^ CFU/mL cell suspensions was taken and 1 mL of xylene was added in the test tube (vortexed for 2 min). This was allowed to stand for 2 h (37 °C) to facilitate the separation of the two phases. Then, the absorbance (600 nm) of the aqueous phase was determined (A).
Cell surface hydrophobicity (%) = [(Ao − A)/Ao] × 100(1)
where Ao = initial absorbance (600 nm) and A = final absorbance.

##### Autoaggregation

The method autoaggregation was followed as described by Vidhyasagar and Jeevaratnam (2013) [21] with slight modification. The cell suspension was incubated at room temperature and the upper layer was evaluated at 0, 2, 4, 6, 10, and 24 h using spectrophotometer at an absorbance of 600 nm. The autoaggregation of the cell surface was estimated using the following equation:Autoaggregation (%) = [(Ao − A)/Ao] × 100(2)
where Ao = initial absorbance (600 nm) and A = final absorbance.

##### Coaggregation Assay

The 2 mL suspension of LAB cell suspension and 1 mL of 5 pathogenic strains, i.e., *Escherichia coli* (MTCC 443), *Bacillus subtilis* (MTCC 10403), *Micrococcus luteus* (MTCC 1809), *Pseudomonas aeruginosa* (MTCC 424), *and Salmonella typhimurium* (MTCC 98), were mixed and incubated (37 °C, 2 h). The absorbance of the combination was measured at 600 nm and monitored. The percentage of coaggregation was computed as follows:Coaggregation (%) = [(A_LAB_ + A_Path_) − A_mix_] × 100/(A_LAB_ + A_Path_)(3)
where A_LAB_ + A_Path_ signifies the absorbance of the LAB–pathogen mixture at time 0 h, and A_mix_ denotes the absorbance of the LAB–pathogen combination at time 2 h.

##### In Vitro Adhesion to Buccal Epithelial Cells

With minor modifications, the methodology described by Somashekaraiah et al. (2019) [22] was utilized to examine the LAB isolates’ ability to attach to buccal epithelial cells in vitro. In brief, buccal epithelial cells were studied for LAB adhesion to their surfaces. A healthy volunteer provided the cells. The epithelial cells were collected and washed twice with saline solution. After that, the cells were centrifuged (5000 rpm, 2 min) and the pellets were collected in saline after discarding the supernatant. A total of 400 µL of diluted buccal epithelial cells (3 × 10^6^ cells) was combined with 100 µL of LAB isolates (10^8^ CFU/mL) and incubated (2 h). The microscopic observation was performed to determine the adherence of LAB to buccal epithelial cells using Gram’s staining [22].

##### HT-29 Cell Culture and Growth Conditions

The adhesion potential of the 6 isolates to human colon cancer cell lines (HT-29) was assessed according to Verhoeckx et al. (2015) [23]. The National Centre for Cell Science in Pune, Maharashtra, India provided HT-29 cell lines (passage #123–130) that were cultured (37 °C, 5% CO_2_) in DMEM (25 mM) without sodium pyruvate but with GluaMAX (Gibco, Paisley, UK). The media were supplemented with 10% (*v*/*v*) FBS (Gibco, UK) and 100 g/mL penicillin and streptomycin. HT-29 cells were subcultured at 1 × 10^5^ cells/mL in a six-well culture plate and cultivated at 37 °C in a humidified CO_2_ atmosphere until they reached 60% confluence in cell medium. On alternate days, the culture medium was changed.

As a part of the adhesion test, isolates were grown (16 h, 37 °C) in MRS broth. After washing twice with PBS, cells were resuspended in DMEM medium at a concentration of 10^8^ CFU/mL. Each well received 1 mL of bacterial suspension, which was incubated for 30 and 60 min at 37 °C (5% CO_2_ atmosphere). The cells were lysed by adding 1 mL of 0.1% Triton-X solution (in PBS) and the non-adherent ones were removed by adding PBS. The solution containing the discharged bacterial cells was serially diluted and plated on MRS agar after 10 min at 37 °C (24 h). The percentage ratio of the initial number of bacteria implanted to that seeded following washing (CFU/mL) was used to determine its adhesion ability. The experiments were carried out in pairs in triplicates [24].

#### 2.3.2. Tolerance Assay

##### Acid and Bile Salt Tolerance

The acid and bile salt tolerances were conducted as described in Wu et al. (2021) [25], with slight modification. The isolated LAB strains (10^8^ CFU/mL) inoculated in 0.3% and 1% ox gall MRS broth (pH 2, 37 °C) for 0, 2, and 4 h were enumerated on MRS agar plates (triplicates) for 24 h at 37 °C. The following formula was used to compute the survival rate (%):Survival rate (%) = [(log CFU N_1_)/(log CFU N_0_)] × 100(4)
where N_1_—total viable count of LAB strains after treatment (CFU/mL), while N_0_—total viable count of LAB strains before treatment (CFU/mL).

##### Simulated Gastric Juice Tolerance Assay

The simulated gastric juice was made by dissolving pepsin (3 g/L of PBS, pH 3; 1:3000 AU/mg, Sisco Research Laboratories Pvt. Ltd., Mumbai, India), and the simulated intestinal juice was made by suspending trypsin (1 g/L of PBS, pH 8; 2000 U/g, Sisco Research Laboratories Pvt. Ltd., India). They were sterilized by passing through a 0.22 μm filter membrane. The isolates when consumed need to tolerate the gastric condition up to 3 h and intestinal conditions up to 8 h as per a normal healthy digestion process. In vitro, using a 5% CO_2_ incubator, the isolates (10^8^ CFU/mL) were inoculated to maintain gastrointestinal conditions. The selected strain’s gastrointestinal tolerance was assessed using viable colony counts [26]. The following equation was used to compute the survival rate:Survival rate% = [(log CFU N_1_)/(log CFU N_0_)] × 100(5)
where N_1_ = total viable count of LAB strains after treatment by simulated gastrointestinal juices and N_0_ = total viable count of LAB strains before treatment [27].

#### 2.3.3. Safety Assessments

##### Antimicrobial Activity

The agar well diffusion method was used to evaluate the antimicrobial activity of the isolates against pathogenic bacteria [28]. *Escherichia coli* MTCC 443, *Bacillus subtilis* MTCC 10403, *Pseudomonas aeruginosa* MTCC 424 and *Salmonella typhimurium* MTCC 98, *Bacillus cereus* MTCC 1272, *Micrococcus luteus* MTCC 1809, *Staphylococcus aureus* MTCC 1144, *Klebsiella pneumonia* MTCC 10309, *Pseudomonus florescens* MTCC 667, and *Klebsiella aerogenes* (*Enterobacter aerognes*) MTCC 2822 were the test organisms. The pathogen (100 µL) was added to Luria Bertani agar plates and spread over the entire surface of the agar plate. Borers were used to create wells on the plates. A total of 100 µL (10^8^ CFU/mL) of overnight grown LAB isolates was transferred into each well (24–48 h, 37 °C). After 18–24 h, the zone of inhibition was measured in millimeters (mm) using the scale.

##### Antibiotic Sensitivity

As per the EFSA (2012) [29], on MRS agar plates, the antibiotic susceptibility of the LAB isolates was tested using the antibiotic disc diffusion method. The MRS agar plates were coated with the LAB isolate culture (10^8^ CFU/mL) and left to dry. After that, the plates were loaded with antibiotic discs and incubated for 24 h at 37 °C. The antibiotic susceptibility pattern of the isolates was determined using 100 µg/discs of streptomycin; 30 µg/discs of chloramphenicol, kanamycin, tetracycline, and vancomycin; 15 µg/discs of azithromycin and erythromycin; 10 µg/discs of gentamicin, ampicillin; 5 µg/discs of rifampicin, methicillin, and cefixime; and 2 µg/disc of clindamycin. The results were interpreted as susceptible, moderate susceptible, or resistance by comparison to performance standards for antimicrobial disc susceptibility tests that describe the interpretative zone diameters [30].

##### Hemolytic Activity

Using the procedure given by Sorokulova et al. (2007) [31] with slight methodology modification, the hemolytic activity of the isolates was tested. The isolates were streak plate inoculated and incubated (37 °C, 48 h) on blood agar plates containing 5% (*w*/*v*) sheep blood. The lysis of red blood cells in the media around the colonies (γ-hemolysis (safe), α-hemolysis, β-hemolysis) was used to test the hemolytic activity of the isolates.

##### DNase Activity

To test for DNase enzyme production, the LAB isolates were streaked onto a deoxyribonuclease (DNase) agar medium. After 48 h (37 °C), the plates were examined for the existence of a DNase activity zone. Positive DNase activity was shown by a pronounced zone around the colonies [32].

### 2.4. Molecular Identification of LAB

Molecular identification of the isolated LAB strains was performed using the universal primers 27F (5′AGAGTTTGATCCTGGCTCAG3′) and 1492R (5′GGTTACCT TGTTACGACTT3′) to amplify 16S DNA sequence, in accordance with the guidelines outlined by Boubezari et al. with certain modification. The isolates, on the basis of their probiotic potential, were subjected to DNA isolation and amplification. The amplified PCR products were sequenced and homology search was conducted using BLAST (Basic Local Alignment Search Tool). The sequences were submitted to the GenBank sequence database and accession numbers were obtained [33,34].

#### Sequencing Homology Search and Phylogenetic Analysis

Using MEGA X, the phylogenetic tree was constructed for the sequenced 16S rRNA region of six LAB isolates from the present study. Maximum likelihood phylogenetic trees with 1000 bootstrap consensus tree were constructed. Tamura–Nei was the best fit model obtained [35]. The initial tree(s) for the heuristic search were automatically generated by applying the neighbor-join and BioNJ algorithms to a matrix of pairwise distances [36].

### 2.5. Antioxidant Assay

The scavenging activity of ABTS radicals was measured following the methodology mentioned by Yang et al. (2020) [37]. Using the method described by Xing et al. (2006) [38], the DPPH radical-scavenging capacity of isolates was assessed.

### 2.6. Preparation of Intact Cells and Intracellular Cell-Free Extracts

The cells of isolates (18 h, 37 °C) were harvested by centrifugation at 2000 rpm for 15 min to obtain the cell-free supernatant (CS), which was filtered (to eliminate bacteria cell debris) through a 0.22 m filter and neutralized (pH 7.4). The pellets obtained after centrifugation were the intact cells (IC), and pellets were suspended in PBS (pH 7.4) and adjusted to 1 × 10^8^ CFU/mL, wherein the cell-free extract (CE) was a sonicated extract (Probe sonicator, ATP-150, Ramson lab equipment, Bangalore, Indian; 15 min at 3 s pulses with 1 min interval in the ice bath) of 1 × 10^8^ CFU/mL cells in PBS (7.4 pH). Then, to remove bacteria cell debris, it was centrifuged at 8000 rpm for 20 min, sterilized (0.22 µm filter), and, finally, the supernatant was collected as CE [39].

### 2.7. Inhibitory Assay for Carbohydrate Hydrolyzing Enzymes

The α-glucosidase inhibition activity was carried out in the same way as before, with minor modifications as described by Kim et al. [40]. The α-amylase inhibition assay was carried out as described in Kwon et al., with minor modification [41]. Intestinal α-glucosidase and pancreatic α-amylase, which are responsible for carbohydrate hydrolysis and absorption, could be inhibited to reduce postprandial hyperglycemia in diabetics [3,42]. A microplate reader was used to measure the absorbance of the reaction of α-glucosidase (405 nm) and α-amylase (540 nm). The following formula is used to calculate the inhibition activity in LAB strains:Inhibition% = (1 − A_S_/A_C_) × 100(6)
where A_S_ = absorbance of the reactants with the sample; A_C_ = absorbance of the reactants without the sample.

### 2.8. Statistical Analysis

All of the tests were performed in triplicate and the results were expressed as mean ± standard deviation. The statistical comparisons between the isolates were performed by one-way analysis of variance (ANOVA), followed by Duncan’s multiple range test using SPSS Software (Version 21.0, Chicago, IL, USA). The results were considered statistically significant if the “*p*” value was 0.05. The graphs were drawn using the GraphPad Prism version 8.0 software (GraphPad Software Inc., San Diego, CA, USA).

## 3. Results

### 3.1. Preliminary Assays of LAB Strains

A total of 40 strains were isolated from the dosa batter and six were identified as *Lactobacillus* spp. according to their phenotypic characterization. All the strains were Gram-positive, catalase negative, and rod-shaped. All the isolates tested in different temperatures had the ability to grow at 37 °C. *Limosilactobacillus fermentum* RAMULAB10 had the capability to tolerate a temperature of 45 °C. The isolates were able to tolerate 2% and 4% salt concentration in the media for their growth. The optimum growth was at pH 7.4, whereas at pH 2, 4, and 6, the growth was mild. Biochemical characterization revealed that the isolates were hetero-fermentative, producing only acid, but no gas from glucose. The isolates were able to ferment glucose, sucrose, and maltose (Table 1). They had the potential to tolerate 0.4% phenol (Table 2). Results obtained had not much difference in the growth at different incubation of 0 h and 24 h with 0.4% phenol; the viable count ranged from 7.30 to 7.89 log CFU/mL. The isolate *Limosilactobacillus fermentum* RAMULAB11 was the most tolerant to 0.4% phenol with 7.89 log CFU/mL viable counts.

### 3.2. Probiotic Properties

#### 3.2.1. Adherence Assay

##### Cell Surface Hydrophobicity

The cell surface hydrophobicity was determined using xylene. Among the isolates, *Lacticaseibacillus casei* RAMULAB08 showed a maximum hydrophobicity of 72.58%. Likewise, *Limosilactobacillus fermentum* RAMULAB10 with 56.99% showed the minimum hydrophobicity (Table 3).

##### Autoaggregation and Coaggregation Assay

Autoaggregation of probiotics is essential for bacterial colonization and protection. At 24 h, all of the isolates had autoaggregation activity greater than 75% (Figure 1A). The strain’s autoaggregation increased as the incubation time increased. Among these probiotic strains, *Limosilactobacillus fermentum* RAMULAB09 showed the highest autoaggregation percentage of 88.50% at 24 h. The coaggregation of the pathogens with the strains could express a moderate ability of coaggregation. The higher coaggregation ability was observed in *Limosilactobacillus fermentum* RAMULAB10. All the isolates could highly coaggregate with *Micrococcusluteus*. The coaggregation of the isolates is shown in Figure 1B.

##### Adhesion Ability to Buccal Epithelial Cell and HT-29 Cell Lines

When isolated buccal epithelial cells were investigated, the adhesion capacity was determined to be 100–200 bacterial cells in each epithelial cell, with the lowest being 24–35 cells. RAMULAB09 had the best adhesion, whereas RAMULAB08 had the lowest (Figure 2) The adherence of the isolates to HT-29 cells was higher than 82% (Table 4). *Lacticaseibacillus casei* RAMULAB07 had the maximum adherence compared with that of the other isolates.

#### 3.2.2. Tolerance Assay

##### Acid Bile Salt Tolerance

The acid and bile tolerance facilitates to assess the survival rate of the isolates at pH 2 with distinct bile concentrations of 0.3% and 1%. Figure 3A,B show the isolates’ ability to survive at an acidic pH 2 with bile tolerance of 0.3% and 1. The survival rate (%) of the LAB isolates was 96% at 0.3% and 95% and above at 1% oxgall concentration and a low pH after incubation for 4 h. At 0.3% bile salt concentration, the survival rate of all six isolates was notably high, as expressed in the graph (Figure 3A,B).

##### Simulated Gastrointestinal Juice Tolerance Assay

The gastrointestinal juice tolerance test of the isolates showed the ability to tolerate the gastric and intestinal conditions. All the six isolates were capable of optimum growth. Later, the growth gradually decreased as the time of incubation increased (Figure 4A,B). The isolate had the most favorable survival rate up to a period of 8 h, and later, the rate decreased gradually.

### 3.3. Safety Assessments

#### 3.3.1. Antimicrobial Activity

The isolates were tested for antimicrobial activity against the pathogenic bacteria. Considerably, the isolates exhibited antimicrobial activity against all the indicator strains. The zone of inhibition scale ranges from 6–20 mm. All the isolates showed a good antimicrobial activity against *M. luteus* and *P. aeruginosa*, which are the opportunistic pathogens. Minimal inhibitory activity was expressed against *B. cereus* and *K. pneumonia* (Table 5). *Lactobacillus* strains have the capability of producing the bacteriocin which could be the reason for the observed inhibitory activity.

#### 3.3.2. Antibiotic Sensitivity

To determine the antibiotic resistance profile, the isolates were tested against 13 antibiotics. The reference standard chart was compared to obtain the results. Evaluation of the probiotic properties’ sensitivity to antibiotics is one of the standards. The six isolates demonstrated sensitivity to chloramphenicol, gentamicin, clindamycin, ampicillin, tetracycline, erythromycin, streptomycin, rifampicin, and azithromycin. Resistance was observed against kanamycin, vancomycin, methicillin, and cefixime (Table 6 and Table 7).

#### 3.3.3. Hemolytic and DNAse Assay

After incubation for 48 h (37 °C), the six LAB isolates showed no zone around the colonies and therefore were considered as safe and classified as γ-hemolysis. The appearance of a clear zone indicates that the isolates were safe to utilize. DNAse activity was also a measure to evaluate the safety of the probiotic formulation. The isolates that demonstrated no zone of inhibition confirmed their non-pathogenic character.

### 3.4. Molecular Identification of LAB

The six biochemically characterized isolates of LAB were amplified to give a sequence length that varied from 730–1348 bp. The homology search of the sequences RAMULAB09, RAMULAB10, RAMULAB11, and RAMULAB12 had >95% similarity to *Limolactobacillus fermentum.* The samples RAMULAB07 and 08 had a similarity of >95% to *Lacticaseibacillus casei*, thus validating the isolates sequenced (Figure 5). The NCBI GenBank accession number for all the isolates is given in Table 8.

### 3.5. Antioxidant Assay

The scavenging activity for ABTS radicals exhibited by the isolates ranged from 20.77% to 89.75%. At 10^3^ CFU/mL cells, RAMULAB07 showed the highest scavenging activity, whereas RAMULAB12 showed the lowest (Figure 6A). The isolates expressed a higher scavenging activity as evaluated by DPPH free radical as the number of cell increased in CFU/mL (Figure 6B). At 10^9^ CFU/mL, RAMULAB10 had the highest radical-scavenging activity (73.23%), followed by RAMULAB09, RAMULAB11, RAMULAB07, RAMULAB08, and RAMULAB12.

### 3.6. Inhibitory Assay for the Carbohydrate Hydrolyzing Enzymes (α-Glucosidase and α-Amylase)

In our study, the inhibitory activity against α-glucosidase and α-amylase was performed using the CS, CE, and IC of the six isolates. CS and CE had a high inhibitory effect on both α-glucosidase and α-amylase for all the isolates. The α-glucosidase inhibition of the isolates for CS, CE, and IC ranged from 7.50 to 65.01%, whereas for α-amylase the inhibition ranged from 20.21 to 56.91%. The intact cells showed the least inhibition compared with that of the supernatant and pellets (Figure 7A,B).

## 4. Discussion

The goal of this study was to obtain probiotic bacteria from fermented batter that could survive the digestive process and inhibit carbohydrate hydrolysis enzymes (α-glucosidase and α-amylase). Dosa batter was discovered to be a rich source of probiotic bacteria with phenotypic features. Six isolates out of 40 made it through the lactic acid bacteria screening. The isolates were tested for viability at various temperatures, salt tolerance, and acid bile tolerance [32]. Food digestion takes up to 2–3 h in the stomach (pH 2) and 3–8 h in the intestine (pH 8), and bile and gastrointestinal tolerance are specific assays to determine the potential of probiotic microorganisms [43,44]. In a certain study, the survival rate (%) of the LAB isolates was >96% at 0.3% and >95% at 1% oxgall concentration at low pH after incubation for 4 h [44]. Similarly, the food isolates strains reported by Montserrat Castorena-Alba et al. showed higher survival rates [45]. Probiotics must survive passage through the gastrointestinal tract in order to maintain and exert their health-promoting effects. All the isolates had the most favorable viability rate in our study, and the survival rate was above 98%. Wang et al. found that 69% and Aziz et al. found that >85% of LAB species survived after being exposed to gastrointestinal juice in a comparative study [46,47]. In our study, the survival rate was consistent, and tolerance to bile acids was comparatively better than the former studies. Certain digestion processes release hazardous metabolites, such as phenol, that are produced by gastrointestinal microbes [48]. Many studies have found that the *Lactobacillus* spp. expressing phenol tolerance have shown a high viability rate [24,49,50]. In this study, the isolate *Limosilactobacillus fermentum* RAMULAB11 was the most tolerant to 0.4% phenol, with 7.89 log CFU/mL viable counts. Previously, *Limosilactobacillus fermentum* NRAMJ5 and *Lactobacillus gastricus* NRAMJ2, reported by Soliman et al. (2021) [50], expressed good phenol tolerance. In this investigation, the isolates were found to be quite efficient in tolerating phenol and assisting them in surviving along the gastrointestinal tract’s passage.

Bacterial adhesion to the human intestinal layer is a complex phenomenon involving many factors such as bacterial and human cell charges, hydrophobicity, polysaccharide (extracellular), and proteins (cell surface). Bacterial cells must overcome all of these barriers in order to achieve proper, strong, or irreversible adhesion [51]. With their hydrophobicity, autoaggregation, and coaggregation abilities, probiotics can help prevent pathogen colonization. The autoaggregation and hydrophobicity of the colonies of the same group of bacteria enable the microorganisms to attach to the gut layer. Because coaggregation is the intercellular adhesion of distinct strains, the probiotic microbe can adhere to the pathogen intercellularly. The autoaggregation and coaggregation of the isolates in this investigation are both high, with autoaggregation exceeding 75% and coaggregation varying. This occurrence aids in the preservation of the intestine’s healthy environment [8,34,52,53].

The isolates are probiotic, and they are generally recognized to be safe (GRAS), yet as per 2002 FAO/WHO guidelines it is recommended to assess the safety of potential probiotics by minimal tests which include antibiotics resistance patterns [54]. Furthermore, antibiotic resistance is assessed in order to check the growth of LABs in the presence of antibiotics and express probiotic potential. Kanamycin, vancomycin, methicillin, and cefixime resistance was found in all six isolates. The zone generated determines the interpretation of the resistance and sensitivity profile, which is based on the CLSI 2018 scale [30,55].

Another significant element for maintaining a healthy microbial balance in the gastrointestinal system is the strain’s antibacterial action against infections. In the current study, all the isolates showed a good antimicrobial activity against *M. luteus* and *Pseudomonas aeruginosa*, which are opportunistic pathogens. In addition, minimal inhibitory activity was expressed against *B. cereus* and *K. pneumonia*. In addition, the *Lactobacillus plantarum* UBLP40 isolated from fermented food possessed antimicrobial activity against *M. luteus*, *S. aureus*, *P. aeruginosa*, and *E. coli* [56]. Plantaricin NC8, produced from *Lactobacillus plantarum* ZJ316, induced membrane permeabilization and cell membrane rupture in *M. luteus* [57]. Similarly, Varma et al. studied the co-incubation with *L. fermentum* strains used to assess the suppression of *S. aureus* and *P. aeruginosa* growth [58]. In the current study, all the isolates showed a good antimicrobial activity against *M. luteus* and *Pseudomonas aeruginosa*, which are opportunistic pathogens. In addition, minimal inhibitory activity was expressed against *B. cereus* and *K. pneumonia*.

Diabetes pathogenesis has been associated with excessive generation of free radicals [59]. Hydroxyl and related radicals, which cause oxidative damage to biomolecules, are the most harmful reactive oxygen species [60]. The bacteria’s cell wall constituents are linked to their ability to scavenge free radicals in intact cells. Antioxidant compounds revealed in LAB strains include NADH, NADPH, antioxidant enzymes, Mn^2+^, bioactive molecules, and exopolysaccharides [61,62]. In this study, isolates have exhibited good antioxidant capability, which is clearly demonstrated by increase in CFU/mL, increasing the ability to scavenging activity of the radical.

The main objective of this study was to evaluate α-glucosidase and α-amylase inhibitory activity of the probiotic strains. The α-glucosidase inhibition of the isolates for CS, CE, and IC ranged from 7.50 to 65.01%, whereas for α-amylase inhibition, it ranged 20.21 to 56.91%. When compared to the commercially available LAB, Son et al. found that the *L. brevis* KU15006 α-glucosidase inhibitory activity of 24.11% for cell-free supernatant (CFS) and 10.563% for cell-free extract (CFE) were the greatest [26]. The *Lactobacillus* strains isolated from yogurt by Ramchandran et al. showed an inhibition higher than 66% [63]. According to Se Young Kwun et al. (2020) [64], the α-glucosidase inhibitory activity of MBEL1397 isolated from kimchi was 3.91%, which is around 2.3 times higher than that of acarbose. The inhibition of intestinal α-glucosidase and pancreatic α-amylase inhibition aids in the treatment of post-prandial hyperglycemia and the regulation of blood glucose levels. The inhibitory activity of α-glucosidase and α-amylase can help to reduce diabetes problems, and probiotic LAB can improve the overall health [2,4,5,63]. According to the investigation by Chen et al. (2014) [39], the intact cells were unable to enter the blood and the CFEs were absorbed from the small intestine into the blood. The fermented beetroot isolated Lactobacillus spp. obtained from fermented beetroot isolates processed for CS and CE expressed a good inhibition against α-glucosidase and α-amylase, suggesting its role in the reduction of hyperglycemia [65]. In this study, the CS and CE have a comparatively higher inhibitory effect than the IC, thus suggesting the presence of inhibitory factors in cell-free supernatant and extract but the least inhibitory factors in intact cell (cell wall not disrupted) of the isolate. The *Lactobacillus* spp. isolated from food source expressed promising results for the inhibition of α-glucosidase and α-amylase. These isolates not only can enhance the gut health but can also reduce the hyperglycemia.

## 5. Conclusions

Metabolic disorders such as diabetes are rapidly increasing with Western eating habits and are becoming more serious. In the present study, an attempt was made to evaluate the beneficiary properties of probiotics isolated from dosa batter for the treatment of diabetes. This study constitutes the first investigation of α-glucosidase and α-amylase inhibitory activity of probiotic LAB from this source. The study revealed a remarkable bile salt and acid tolerance, gastrointestinal tolerance, autoaggregation and coaggregation abilities, hydrophobicity, and antibiotic and antimicrobial properties. All the six isolates displayed substantial inhibitory activity of α-glucosidase and α-amylase when tested using their IC, CS, and CE, with IC faring to be the most optimal. The results are promising in suggesting the beneficiary potential of the probiotic LAB isolated from dosa batter.

## Figures and Tables

**Figure 1 microorganisms-10-01195-f001:**
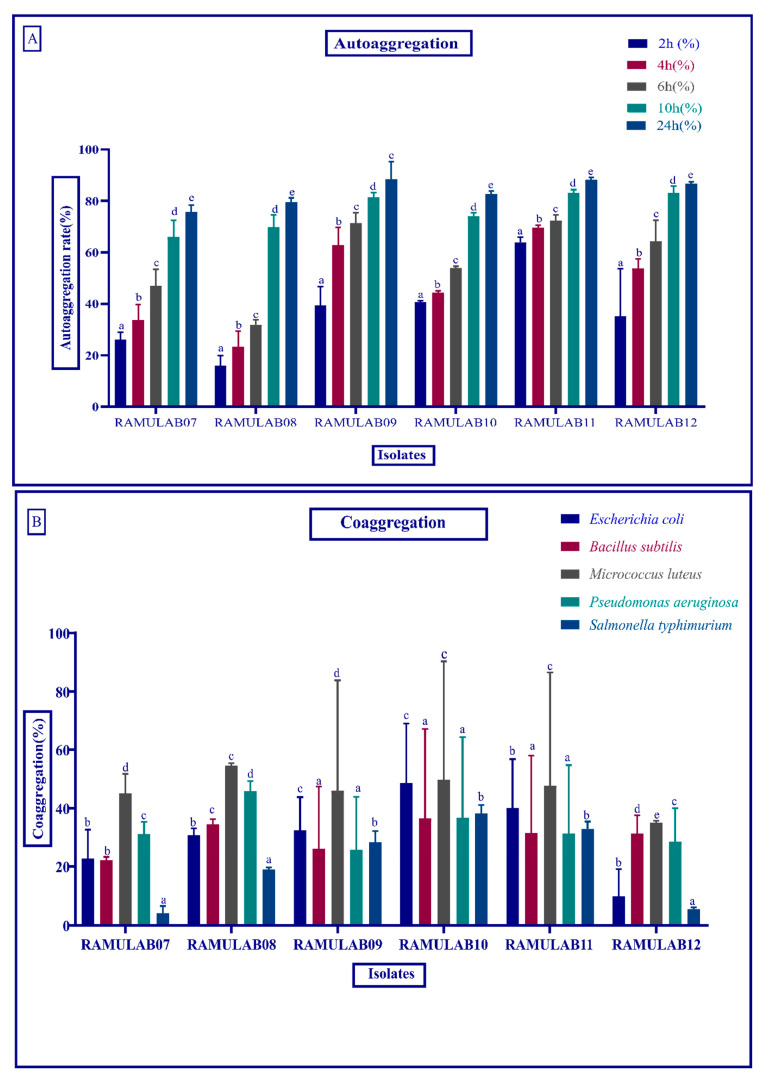
The (**A**) autoaggregation (%) strains at different time interval at room temperature and (**B**) coaggregation (%) of LAB strains after incubation of 2 h at room temperature. Data are expressed as the mean ± SD. Means in aggregation for 2 h with distinct superscripts (a–e) are significantly different (*p* ≤ 0.05), as separated by Duncan multiple range test.

**Figure 2 microorganisms-10-01195-f002:**
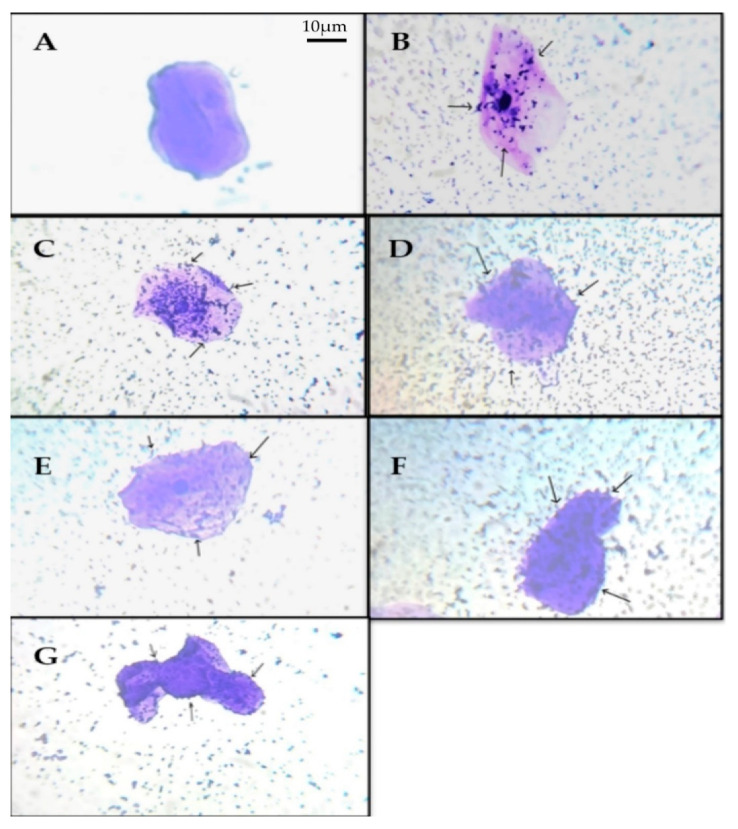
LAB strains adhesion to buccal epithelial cells observed under a light microscope. (**A**) Buccal epithelial cells (control). (**B**) The adhesion of isolate: RAMULAB07 (**B**), RAMULAB08 (**C**), RAMULAB09 (**D**), RAMULAB10 (**E**), RAMULAB11 (**F**), and RAMULAB12 (**G**) to buccal epithelial cells. Note: the black arrow shows the LAB strains attached to the epithelial cells.

**Figure 3 microorganisms-10-01195-f003:**
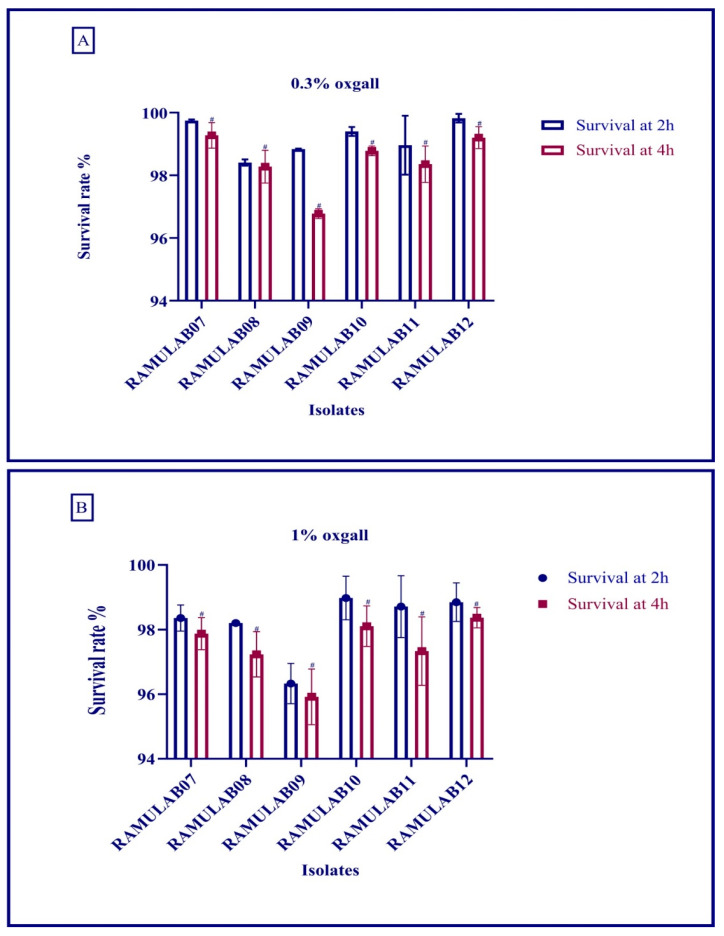
Acid and bile survival rate of LAB strains isolated from dosa batter sample with an acidic pH 2 value and under different bile salt conditions: (**A**) 0.3%, (**B**) 1% bile salt concentration conditions for 2 and 4 h at 37 °C in MRS agar plates. Data are expressed as the mean ± SD. Means in survival rate with time interval of 2 h with superscripts (#) are significantly different (*p* ≤ 0.05), as separated by Duncan multiple range test.

**Figure 4 microorganisms-10-01195-f004:**
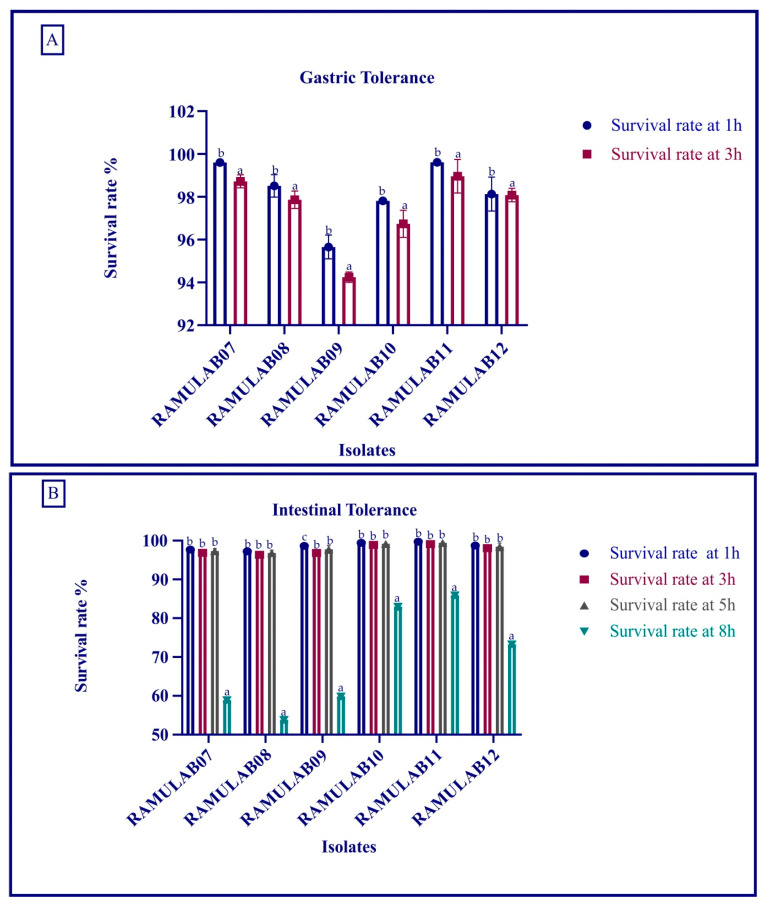
Survival rate of isolates in gastric (**A**) and intestinal (**B**) juice. Data are expressed as the mean ± SD. Means in survival rate for time interval (1 h, 3 h, 5 h, and 8 h) with distinct superscripts (a–c) are significantly different (*p* ≤ 0.05), as separated by Duncan multiple range test.

**Figure 5 microorganisms-10-01195-f005:**
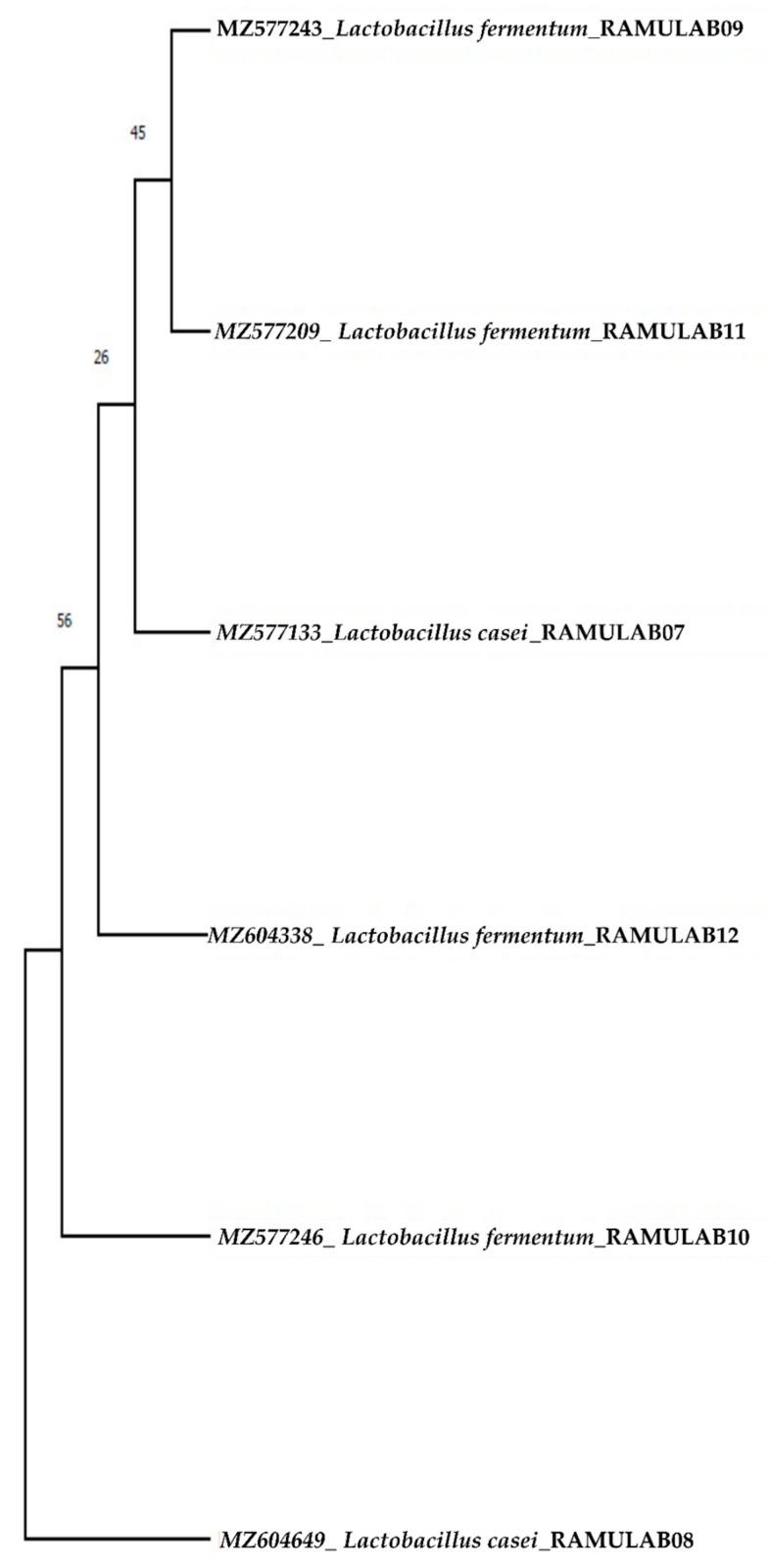
Phylogenetic tree of LAB isolates (RAMULAB07–12) from dosa batter samples based on maximum likelihood bootstrap analysis of 16S rDNA.

**Figure 6 microorganisms-10-01195-f006:**
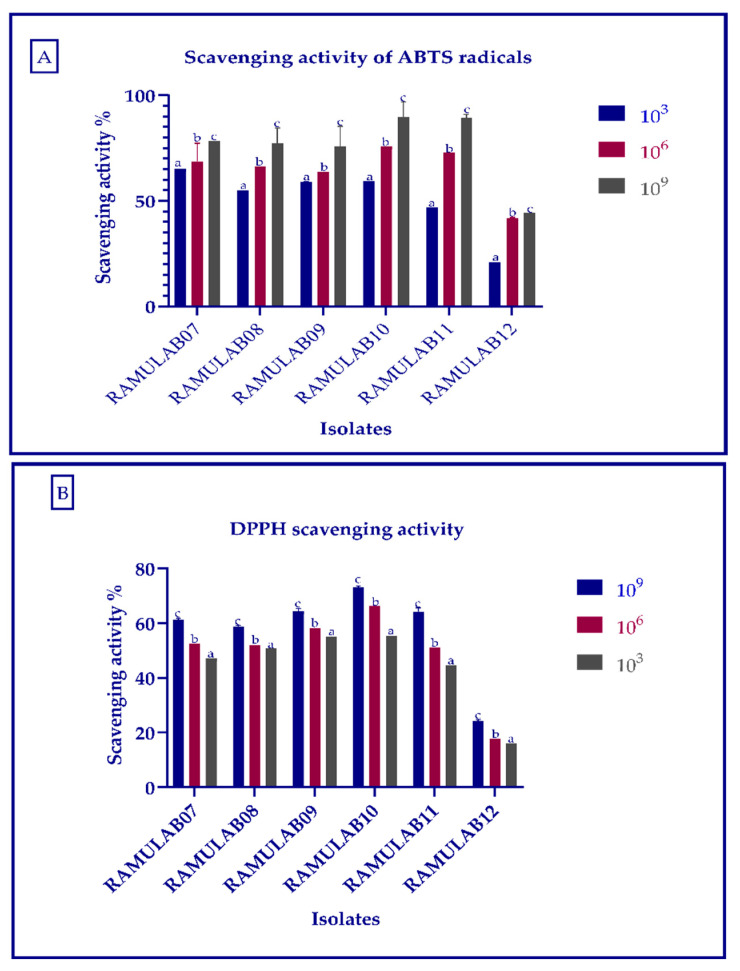
The (**A**) scavenging activity of ABTS radicals and (**B**) DPPH free radical scavenging activity of the isolates. Data are expressed as the mean ± SD. Means in scavenging activity of different CFU/mL with distinct superscripts (a–c) are significantly different (*p* ≤ 0.05), as separated by Duncan multiple range test.

**Figure 7 microorganisms-10-01195-f007:**
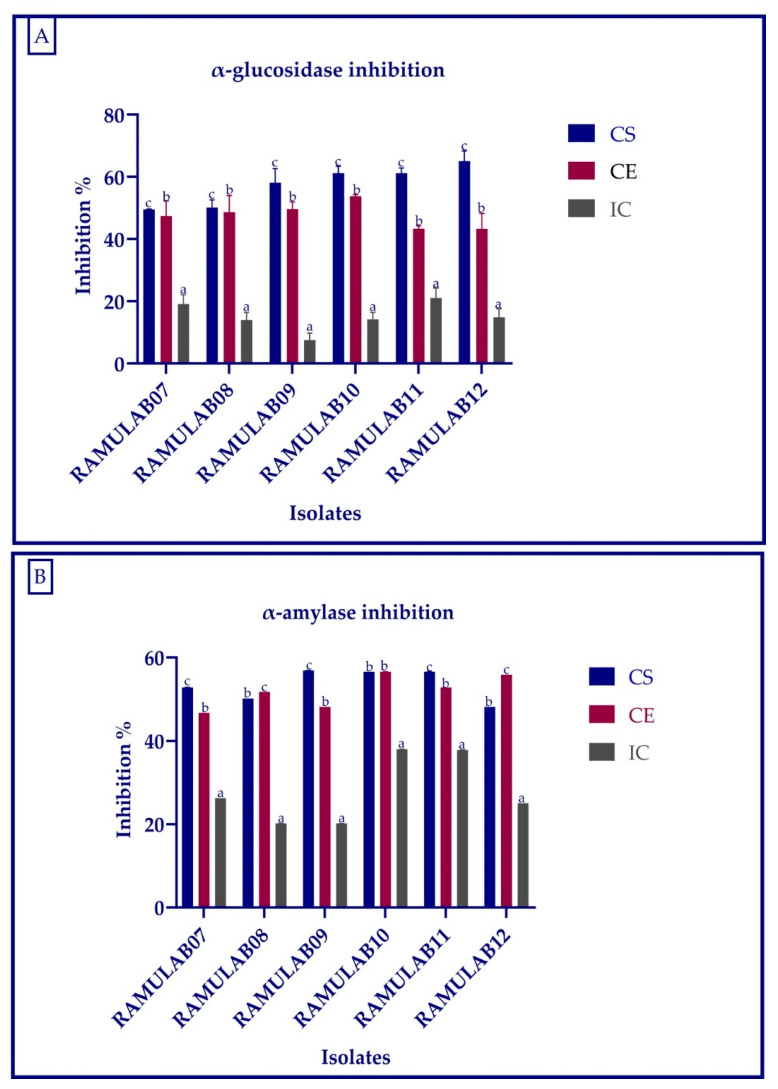
α-glucosidase (**A**) and α-alpha amylase (**B**) inhibitory activity of the isolates. Data are expressed as the mean ± SD. Means in inhibition activity of CS, CE, and IC with distinct superscripts (a–c) are significantly different (*p* ≤ 0.05), as separated by Duncan multiple range test.

**Table 1 microorganisms-10-01195-t001:** LAB strains isolated from dosa batter sample were studied for their phenotypic characteristics and fermentation ability.

	Isolates *					
Tests	RAMULAB07	RAMULAB08	RAMULAB09	RAMULAB10	RAMULAB11	RAMULAB12
Gram staining	Positive					
Catalase	−	−	−	−	−	−
Morphology	Rod	Rod	Rod	Short Rod	Rod	Rod
Growth at Different Temperature (°C)						
4	−	−	−	−	−	−
10	−	−	−	−	−	−
37	+	+	+	+	+	+
45	−	−	−	+	−	−
50	−	−	−	−	−	−
Growth at Different NaCl Concentration						
2%	+	+	+	+	+	+
4%	+	+	+	+	+	+
7%	−	−	−	−	−	−
10%	−	−	−	−	−	−
Carbohydrates Fermentation						
Glucose	+	+	+	+	+	+
D-xylose	−	−	+	−	+	+
L-xylose	−	−	−	−	−	−
Sucrose	+	+	+	+	+	+
Mannitol	−	+	−	+	+	+
Maltose	+	+	+	+	+	+
Lactose	−	+	−	+	+	+
Galactose	−	+	−	+	+	+
Arabinose	−	−	−	−	−	−
Starch	−	−	−	−	−	−
Growth at Different pH						
2	+	+	+	+	+	+
4	+	+	+	+	+	+
6	+	+	+	+	+	+
7.4	+	+	+	+	+	+

* “+” indicates presence; “−” indicates absence.

**Table 2 microorganisms-10-01195-t002:** The phenol tolerance of the isolates.

	Phenol Tolerance (Log CFU/mL) *	
Isolates	0 h	24 h
*Lacticaseibacillus casei* RAMULAB07	7.44 ± 0.10 ^c^	7.34 ± 0.12 ^a^
*Lacticaseibacillus casei* RAMULAB08	7.42 ± 0.08 ^c^	7.30 ± 0.08 ^a^
*Limosilactobacillus fermentum* RAMULAB09	7.38 ± 0.03 ^b^	7.64 ± 0.55 ^c^
*Limosilactobacillus fermentum* RAMULAB10	7.30 ± 0.22 ^a^	7.53 ± 0.80 ^b^
*Limosilactobacillus fermentum* RAMULAB11	7.67 ± 0.05 ^d^	7.89 ± 0.60 ^d^
*Limosilactobacillus fermentum* RAMULAB12	7.45 ± 0.09 ^c^	7.69 ± 0.36 ^c^

* Values are reported as mean ± SD. Means in the same column indicated by different letters (a–d) are significantly different (*p* ≤ 0.05), as separated by Duncan’s multiple range test.

**Table 3 microorganisms-10-01195-t003:** The cell surface hydrophobicity of the isolates.

Isolates	Cell Surface Hydrophobicity (%) *
*Lacticaseibacillus casei* RAMULAB07	65.71 ± 3.91 ^b^
*Lacticaseibacillus casei* RAMULAB08	72.58 ± 6.06 ^e^
*Limosilactobacillus fermentum* RAMULAB09	65.30 ± 0.20 ^b^
*Limosilactobacillus fermentum* RAMULAB10	56.99 ± 2.04 ^a^
*Limosilactobacillus fermentum* RAMULAB11	67.20 ± 2.64 ^c^
*Limosilactobacillus fermentum* RAMULAB12	69.31 ± 0.18 ^d^

* Values are reported as mean ± SD. Means in the same column indicated by different letters (a–e) are significantly different (*p* ≤ 0.05), as separated by Duncan’s multiple range test.

**Table 4 microorganisms-10-01195-t004:** Adhesion expressed in percentage of the isolates adhering to HT-29 cells.

Isolate	Adhesion (%) *
*Lacticaseibacillus casei* RAMULAB07	88.34 ± 06.12 ^d,e^
*Lacticaseibacillus casei* RAMULAB08	84.86 ± 15.59 ^c^
*Limosilactobacillus fermentum* RAMULAB09	87.32 ± 02.01 ^d^
*Limosilactobacillus fermentum* RAMULAB10	83.61 ± 04.39 ^b^
*Limosilactobacillus fermentum* RAMULAB11	82.41 ± 11.28 ^a^
*Limosilactobacillus fermentum* RAMULAB12	87.71 ± 01.62 ^d^

* Values are reported as mean ± SD. Means in the same column indicated by different letters (a–e) are significantly different (*p* ≤ 0.05), as separated by Duncan’s multiple range test.

**Table 5 microorganisms-10-01195-t005:** Antimicrobial activity of isolates.

Isolates	Pathogens *									
	*B. cereus*	*S. aureus*	*S. typhimurium*	*E. coli*	*P. aeruginosa*	*K. pneumoniae*	*M. luteus*	*B. subtilis*	*P. florescens*	*K. aerogenes*
*Lacticaseibacillus casei* RAMULAB07	*+*	*++*	*−*	*++*	*+++*	*+*	*+++*	*++*	*++*	*−*
*Lacticaseibacillus casei* RAMULAB08	*−*	*++*	*+*	*+*	*++*	*+*	*+++*	*+*	*++*	*+*
*Limosilactobacillus fermentum* RAMULAB09	*+*	*++*	*+*	*+*	*+++*	*−*	*+++*	*++*	*++*	*+*
*Limosilactobacillus fermentum* RAMULAB10	*+*	*++*	*+*	*+*	*+++*	*−*	*+++*	*+*	*++*	*+*
*Limosilactobacillus fermentum* RAMULAB11	*−*	*++*	*−*	*+*	*++*	*−*	*+++*	*−*	*++*	*−*
*Limosilactobacillus fermentum* RAMULAB12	*−*	*++*	*−*	*+*	*++*	*−*	*+++*	*−*	*++*	*−*

* Symbols show zones of inhibition (mm): (−): no inhibition; (+): weak (<7); (++): good (9–15); (+++): strong (>15).

**Table 6 microorganisms-10-01195-t006:** Breakpoints of sensitivity/resistance in the mm inhibitory zone of respective antibiotics.

Sl. No.	Antibiotic	The Inhibitory Zone (S/R mm)
1	Chloramphenicol (C)	(≥18/≤12)
2	Gentamicin (GEN)	(≥15/≤12)
3	Clindamycin (CD)	(≥19/≤14)
4	Ampicillin (AMP)	(≥17/≤14)
5	Kanamycin (K)	(≥18/≤12)
6	Tetracycline (TET)	(≥19/≤14)
7	Vancomycin (V)	(≥17/≤14)
8	Erythromycin (E)	(≥23/≤13)
9	Streptomycin (STR)	(≥15/≤12)
10	Rifampicin (RIF)	(≥20/≤16)
11	Methicillin (MET)	(≥22/≤4)
12	Azithromycin (AZM)	(≥13/≤12)
13	Cefixime (CEF)	(≥21/≤2)

**Table 7 microorganisms-10-01195-t007:** Antibiotic susceptibility test of the isolates representing resistance and sensitivity based on Clinical Laboratory Standards Institute (CLSI), 2018 [30]. Chloramphenicol (C), gentamicin (GEN), clindamycin (CD), ampicillin (AMP), kanamycin (K), tetracycline (TET), vancomycin (V), erythromycin (E), streptomycin (STR), rifampicin (RIF), methicillin (MET), azithromycin (AZM), and cefixime (CEF).

Isolates *	C	GEN	CD	AMP	K	TET	VA	E	STR	RIF	MET	AZM	CEF
*Lacticaseibacillus casei* RAMULAB07	S	S	S	S	R	S	R	S	S	S	R	S	S
*Lacticaseibacillus casei* RAMULAB08	S	S	S	S	R	S	R	S	S	S	R	S	S
*Limosilactobacillus fermentum* RAMULAB09	S	S	S	S	R	S	R	S	S	S	R	S	S
*Limosilactobacillus fermentum* RAMULAB10	S	S	S	S	R	S	R	S	S	S	R	S	S
*Limosilactobacillus fermentum* RAMULAB11	S	S	S	S	R	S	R	S	S	S	R	S	S
*Limosilactobacillus fermentum* RAMULAB12	S	S	S	S	R	S	R	S	S	S	R	S	S

* “S” indicates sensitivity; “R” indicates resistance.

**Table 8 microorganisms-10-01195-t008:** The NCBI GenBank accession number of the isolates and the organism.

Isolates	Accession Number	Organism
RAMULAB07	MZ577133	*Lacticaseibacillus casei*
RAMULAB08	MZ604649	*Lacticaseibacillus casei*
RAMULAB09	MZ577243	*Limolactobacillus fermentum*
RAMULAB10	MZ577246	*Limolactobacillus fermentum*
RAMULAB11	MZ577209	*Limolactobacillus fermentum*
RAMULAB12	MZ604338	*Limolactobacillus fermentum*

## Data Availability

The datasets presented in this study can be found in online repositories. The names of the repository/repositories and accession number(s) can be found in the article.

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
