# Peer review of "Probiotic Potential Lacticaseibacillus casei and Limosilactobacillus fermentum Strains Isolated from Dosa Batter Inhibit α-Glucosidase and α-Amylase Enzymes"

_microorganisms, 2022, doi:10.3390/microorganisms10061195_

Round 1

Reviewer 1 Report

The article presents a set of analyzes to evaluate the probiotic properties of lactic strains isolated from a traditional fermented food, the dosa batter. The results presented underline the probiotic potential of the isolated lactic strains, in particular their ability to inhibit enzymes involved in the degradation of sugars. This result sheds light on the antidiabetic potential of consuming this traditional fermented food.

I have a few remarks on the manuscript:
- The inhibition effects against the α-glucosidase and α-amylase enzymes are contained in the supernatant, would it not be relevant to test these inhibitory effects in the supernatant extracted from this food?
- For the bile acid resistance test, how do you explain that the strains have better resistance to a 1% concentration of bile acids compared to a 0.3% concentration?
- There is an error in the legends of figure 4 (@ instead of at)

Author Response

Response to reviewer comments:

All the authors thank the reviewer for their thoughtful suggestions to improve the quality of the manuscript. The response to all the comments point-by-point are provided below.

Reviewer 1 comments

The article presents a set of analyzes to evaluate the probiotic properties of lactic strains isolated from a traditional fermented food, the dosa batter. The results presented underline the probiotic potential of the isolated lactic strains, in particular their ability to inhibit enzymes involved in the degradation of sugars. This result sheds light on the antidiabetic potential of consuming this traditional fermented food.

I have a few remarks on the manuscript:

  1. The inhibition effects against the α-glucosidase and α-amylase enzymes are contained in the supernatant, would it not be relevant to test these inhibitory effects in the supernatant extracted from this food?

Authors’ response: The authors appreciate the reviewer for his valuable suggestion on testing the whole supernatant of the extracted food. The authors would like to mention that in the current study Lacticaseibacillus casei and Limosilactobacillus fermentum were isolated and characterized for the first time and the same was deposited. In this regard, special emphasis on antidiabetic attributes were investigated for the above two organisms (the crux of the paper). Also, we would like to mention that since dosa batter harbours not just the Lactobacillus spp. but also Leuconostoc spp., Streptococcus spp. and Bacillus spp. determining the inhibitory effect of the supernatant extracted from this food would be the metabolites released from all these the organisms in combination. In this study, the inhibition effects against the α-glucosidase and α-amylase enzymes are mainly focused on the abundantly present Lactobacillus spp. (cell-free supernatant (CS), cell-free extract (CE), and intact cell (IC)) isolated from the Dosa batter. However, the authors will consider the insight suggestions provided by the reviewer and further continue in the same line in the near future.

  1. For the bile acid resistance test, how do you explain that the strains have better resistance to a 1% concentration of bile acids compared to a 0.3% concentration?

Authors’ response: The authors would like to mention that there was a typological error in figure 3. Hence the figure 3 has been remodeled to avoid further confusions. All the isolates had high survival rate at 0.3% bile salt concentration (pH 2) compared to a 1% concentration of bile acids.

  1. There is an error in the legends of figure 4 (@ instead of at)

Authors’ response: According to the reviewer’s suggestion, the error in the legends of figure 4 @ was replaced with “at” and the figure was replaced.

Reviewer 2 Report

You can see the review in the attached file.

Author Response

Response to reviewer comments:

All the authors thank the reviewer for their thoughtful suggestions to improve the quality of the manuscript. The response to all the comments point-by-point are provided below.

Reviewer 2 comments

The article entitled “Probiotic potential Lactobacillus casei and Limosilactobacillus fermentum strains isolated from dosa batter inhibit α-glucosidase and α-amylase enzymes” (Manuscript ID microorganisms-1729741) is very interesting and presents a huge work, but it requires a profound revision and correction before being published. The manuscript requires also an extensive editing of English language and style and checking by a native.

Authors’ response: The authors thank the reviewer and as per his suggestion, the entire manuscript has been proofread by a native English speaker to minimize/correct the typological and grammatical errors in the manuscript

Major comments:

  1. Methods are poorly or wrongly described. They do not provide sufficient details to allow others to replicate the experiments. The formulas used in the methods are wrong!

Authors’ response: According to the reviewer’s suggestion, the entire methodology section has been remodeled to provide sufficient details and allow others to replicate the experiments. Also, the formulas in the methodology section have been corrected to be more precise.

  1. Discussion is very poor. Authors should discuss the results and write how they can be interpreted (are obtained result good or no, satisfactory, have beneficial potential? etc), also from the perspective of previous studies and of the working hypotheses. The implications of the results should be highlighted!

Authors’ response: According to the reviewer’s suggestion, the entire discussion has been remodeled for better understanding and to be more precise. Also, the results are discussed from the perspective of previous studies and of the working hypotheses to improve the quality of the manuscript. Finally the implication of the results of the current study has been highlighted.

  1. I suggest using the new taxonomic names (valid from 2020) like Limosilactobacillus fermentum, and Lacticaseibacillus casei

Authors’ response: According to the reviewer’s suggestion, the new taxonomic names were replaced all through the manuscript.

  1. Authors should provide data about model and producer of laboratory device used.

Authors’ response: According to the reviewer’s suggestion, the model and producer of laboratory device used were added in chapter 2.1 and mentioned wherever necessary in the manuscript.

  1. Authors should provide the composition of used microbial media or name of producer and catalogue number of the medium.

Authors’ response: According to the reviewer’s suggestion, the microbial media producer has been added in chapter 2.1 in the manuscript and the catalogue number of MRS medium has been added.

  1. Lack of results of statistical analysis. Which differences of results are statically significant?

Authors’ response: The authors would like to mention that the data obtained in the present study are the mean of triplicate determinations expressed as mean ± SD. Hence, to avoid confusions and to be precise statistical significance has been deleted in the entire manuscript. Also, all the graphs were drawn using Graphpad prism version 8.0 and the same has been mentioned in the manuscript.

  1. Why the Authors did not used the journal template? For example the manuscript lines are not numbered… equations are not numbered….

Authors’ response: According to the reviewer’s suggestion, the manuscript lines are numbered and equations are numbered.

  1. Authors should check the bibliography, in some positions the abbreviated journal name is lacking, in others the “strange” numbers are between journal name and year of publication

Authors’ response: According to the reviewer’s suggestion, the bibliography was thoroughly checked and the references were reframed according to the journal format.

Minor remarks:

  1. Correct Celsius degree, is °C not 0C

Authors’ response: According to the reviewer’s suggestion, “°C” were replaced in the place of “0C”.

  1. When centrifugation parameters are described Authors should provide data as ... x g or provide date on rotor diameter and centrifuge model.

Authors’ response: According to the reviewer’s suggestion, the centrifuge model information has been provided in chapter 2.1.

  1. A lot of spaces are missed or unnecessary, also Italic font is missed or unnecessary.

Authors’ response: According to the reviewer’s suggestion, the manuscript was thoroughly checked for spaces and itaclic font and necessary changes were done.

  1. Unify in the text and figures: coaggregation or co-aggregation

Authors’ response: According to the reviewer’s suggestion, the word “coaggregation” was unified throughout the manuscript.

  1. Reference numbers should be placed in square brackets [ ], please add the number just after mentioned publication, e.g. “… described in Somashekaraiah et al. (2019) [22] was utilised to….”

Authors’ response: According to the reviewer’s suggestion, the citation number just after mentioned publication were added and replaced in square brackets in the entire manuscript.

  1. The past tense should be used in the whole manuscript (was dome, were used, etc.)

Authors’ response: According to the reviewer’s suggestion, the past tense were used and sentences were reframed as per the suggestions.

Other errors in particular parts of manuscript

Introduction:

  1. Rewrite the sentence: “The nature of fermentation in a majority of fermented foods and traditional Indian dishes is by LAB associated with cereals and legumes [13]”

Authors’ response: According to the reviewer’s suggestion, the sentence was reframed.

Chapter 2.1 (altered to 2.1.1)

  1. For the assays, the isolates was grown on MRS → ….were grown …

Authors’ response: According to the reviewer’s suggestion, the sentence was reframed.

  1. Was the calibration for isolates performed or were values for E. coli assumed (i.e. that 1 McFarland = 3x10^8 cells/mL)? Because it is not the same!

Authors’ response: According to the reviewer’s suggestion, the calibrations for each isolates wereperformed.

Chapter 2.2

  1. salt concentration… ->NaCl concentration …

Authors’ response: According to the reviewer’s suggestion, the word salt was replaced to “NaCl”.

  1. I don’t understand the sentence: “The cell viability for bacterial enumeration (0 h and 24 h ) was performed by the plate count method [19]” → so how you did it? The cited references do not provide the description.

Authors’ response: According to the reviewer’s suggestion, the sentence was reframed to avoid further confusions. The bacterial enumeration was done using plate count method to determine the cell viability and the same has been mentioned in detailed.

Chapter 2.3.1.1.– rewrite it

  1. Separately describe the hydrophobicity determination, then auto-aggregation and finally co-aggregation. Use separate formulas.

Authors’ response: According to the reviewer’s suggestion, the methodology of hydrophobicity determination, then auto-aggregation and finally co-aggregation were written separately with respective formula.

  1. Correct the formulas in the manuscript because now they are wrong!

Aggregation (%) = (Ao-A)/Ao×100 it's not the same as: Aggregation (%) = [(Ao-A)/Ao]×100!!!

Authors’ response: According to the reviewer’s suggestion, the formula (1) was corrected accordingly.

  1. In the sentence: “The preparation of cell suspensions is similar to that used for autoaggregation. 2 mL suspension of different LAB isolates and5 pathogenic strains i.e.,…” - explain how many (volume, optical density?) of these pathogenic strains have been added?

Unify the way of writing formulas and correct the formula:

Coaggregation (%) = [(ALAB + APath)-Amix]x100/ (ALAB + APath)

Authors’ response: According to the reviewer’s suggestion, the sentence was reframed , formula (3) correction and the explanation of the volume of pathogenic strains have been added.

Chapter 2.3.1.3 (altered to Chapter 2.3.1.5)

  1. First sentence: grammar should be improved: The adhesion .... was assessed according to Verhoeckx et al. [23]

Authors’ response: According to the reviewer’s suggestion, the sentence was reframed.

  1. Second sentence: “The National Centre for Cell Science in Pune, Maharashtra, India purchased provided HT-29 cell lines (passage #123-130)…” - is the word ‘purchased’ necessary?

Authors’ response: According to the reviewer’s suggestion, in the sentence the word ‘purchased’ was removed.

  1. “… in a humidified CO2 atmosphere until they reached % confluence in cell medium.” - how many %?

Authors’ response: According to the reviewer’s suggestion, in the sentence of confluence obtained in cell medium was 60%.

  1. “Cells were resuspended in DMEM medium and washed twice with PBS at a concentration of 108 CFU/mL” – it does not make sense. First washing, then resuspending?

Authors’ response: According to the reviewer’s suggestion, the sentence was reframed to avoid further confusions.

Chapter 2.3.2.1.

  1. Correct the formula (see also suggestions below) Survival rate (%) = Log CFU/mL at time (t)/ Log CFU/mL at initial time (0) × 100.

Provide data on enzymes (trypsin, pepsin) used (producer, units of activity etc.)

Authors’ response: According to the reviewer’s suggestion, the formula (4)was corrected the data on enzymes is provided in the chapter 2.1.

Chapter 2.3.2.2.

  1. I don’t understand the sentence: “In a % CO2 incubator, isolates (108 CFU/mL) when consumed need to tolerate the gastric condition up to 3 h and intestinal conditions up to 8 h as per normal healthy digestion process. The selected strain's gastrointestinal tolerance was assessed using viable colony counts [26].” Please, rewrite it.

Survival rate %= (log CFU N1)/(log CFU N0)x100 → this is wrong formula

Should be: Survival rate %= [(log CFU N1)/(log CFU N0)]x100

Authors’ response: According to the reviewer’s suggestion,the sentence was rephrased to avoid further confusion. The formula (5) was corrected accordingly.

  1. I strongly suggest to use the unified mode of writing formulas (not only for chapters 2.3.2.1 and 2.3.2.2.). For example you can write like this:

Survival rate (%) = [(logN1)/(log N0)] × 100,

Where: N1 - Total viable count of LAB strains after treatment by simulated gastrointestinal juices expressed as CFU/mL, while N0 - Total viable count of LAB strains before treatment expressed as CFU/mL.

Authors’ response: According to the reviewer’s suggestion, the formula was corrected accordingly throughout the manuscript.

Chapter 2.3.3.1

  1. “…and Salmonella entericatyphimuriumMTCC 98…” there is not proper name of bacteria. It should be ‘Salmonella Typhimurium’ or ‘Salmonella enterica entericaser. Typhimurium’

Authors’ response: According to the reviewer’s suggestion, the ‘Salmonella typhimurium’ was retained but removing the subs. entericain the sentence.

  1. “The pathogen (100 μL) was added to LB agar (Luria Bertani agar) plates.” – What was the bacteria concentration (CFU/ml)?

Authors’ response: According to the reviewer’s query, the pathogen of volume 100 μL from the overnight (24h) grown LB broth culture was added and spread onto entire agar surface to obtain the uniform distribution of pathogen. The sentence was rephrased to avoid further confusions.

  1. “A 100 μL of overnight grown LAB isolates were transferred….” – what was the LAB concentration in this suspension?

Authors’ response: According to the reviewer’s query, the 100 μL of overnight grown (108 CFU/mL) was transferred into well.

  1. At the end of this chapter add how the antimicrobial activity was evaluated/measured and expressed.

Authors’ response: According to the reviewer’s suggestion,the antimicrobial activity i.e., the inhibition was measured in “mm” was added to the sentence.

Chapter 2.3.3.2

  1. Add “EFSA (2012)” to the bibliography list .Provide the data of antibiotic disks producer. Correct errors (names from big letter, lacking spaces etc)

Authors’ response: According to the reviewer’s suggestion, the bibliography was added and the producer of antibiotic disks was added in chapter 2.1. The minor errors were rectified.

Chapter 2.3.3.2

  1. I suggest change the first sentence into: “Using the procedure given by Sorokulova et al. [30] with slight methodology modification, the hemolytic activity of the isolates was tested.”

Add the name of media supplier and catalogue number or provide their composition.

“… reddish zone…” - are you sure about the zone colour?

Authors’ response: According to the reviewer’s suggestion, the sentence was reframed and the suppliers of the chemicals were added in chapter 2.1. No zone was formed around the streaked isolates indicating γ –hemolysis (result).

Chapter 2.5

  1. With certain changes, the DPPH radical-scavenging capacity of isolates was assessed using Xing et , (2006)[37]. – What it means? Using the method described by Xing et al. (2006) [37]?

Authors’ response: According to the reviewer’s suggestion, the sentence was rephrased to avoid further confusions.

Chapter 2.6

  1. Improve the English language in this chapter or rather rewrite it. Especially the sentences: “The cells of isolates (18 h, 37 0C) harvested by centrifugation at 2000 rpm, 15 min to obtain the cell-free supernatant (CS) was filtered (eliminate bacteria cell debris) through a 0.22 m filter and neutralised (pH 7.4)” and “The cell-free extract (CE), is a sonicated extract (15 min at 3 s pulses with 1 min interval in the ice bath) of 1× 108 CFU/mL cells in PBS (7.4 pH). Then, to remove bacteria cell debris, it was centrifuged at 8000 rpm for 20 min, and sterilized (0.22 μm filter) supernatant was collected as CE”

            Add the model of device use for sonication.

Authors’ response: According to the reviewer’s suggestion, the sentence was improvised to avoid further confusions. The model of sonicator was added.

Results

Chapter 3.1.

  1. I suggest: … catalase negative, and rod-shaped (instead of bacillus).

Authors’ response: According to the reviewer’s suggestion,instead of bacillus the “rod-shaped” word was replaced.

  1. Sentence: “Limosilactobacillus fermentum RAMULAB11 had the capability to tolerate a temperature of 45 °C” does not match the data in the table 1. In Table 1 the RAMULAB10 has such ability, and not RAMULAB11! Check and correct text or table!

Authors’ response: According to the reviewer’s suggestion, the correction was done accordingly to match the results in the table.

  1. “…were hetro-fermentative…” → … were hetero-fermentative…

Authors’ response: According to the reviewer’s suggestion, the word hetro-fermentative was replaced with hetero-fermentative.

  1. “The isolates were able to ferment sucrose, maltose, lactose and galactose (Table 1)” – not all isolates fermented lactose!!

Authors’ response: According to the reviewer’s suggestion, the sentence were rephrased to avoid further confusion.

  1. I suggest change the sentence “Results obtained had no much difference in the growth at different incubation of 0 h and 24 h. With 4 % phenol, the viable count ranged from 7.30 to 7.89 log CFU/mL. into “Results obtained had no much difference in the growth at different incubation of 0 h and 24 h with 0.4 % phenol, the viable count ranged from 7.30 to 7.89 log CFU/mL.”

Authors’ response: According to the reviewer’s suggestion, the punctuation was removed in the sentence to repharse.

  1. “The isolate Limosilactobacillus fermentum RAMULAB11 was the most tolerant to 0.4% phenol with 7.89 log CFU/mL viable counts.” Are you sure? RAMULAB11 initial number was also the highest (7.67 ± 0.05)! Check the data in table 2!

Authors’ response: According to the reviewer’s query,the isolate Limosilactobacillus fermentum RAMULAB11 was the most tolerant to0.4% phenol at oh and 24h.

  1. Below many of the tables the Authors wrote: “values are expressed as Mean ± SD and significance is p ≤0.05”. Please, provide adequate marks/annotations. Are any of presented values statistically different?

Authors’ response: The experiment was done in triplicates and the results are expressed in Mean ± SD and the statistical significance has been removed throughout the manuscript since the results were not compared with the controls to avoid further confusions.

Chapter 3.2.1.1.

  1. “The cell surface hydrophobicity was determined using xylene, among the isolates, Lactobacillus casei RAMULAB08…” I suggest to change it into” “The cell surface hydrophobicity was determined using Among the isolates, Lactobacillus casei RAMULAB08 … RAMULAB09 with 65.30% showed the minimum hydrophobicity (Table 3).”

And, what do you think about RAMULAB10 with 56,99% hydrophobicity?

Authors’ response: According to the reviewer’s suggestion, the sentence was reframed to avoid further confusions.

Table 3

  1. Change the column title into: Cell surface hydrophobicity (%). Are differences statistically significant?

Authors’ response: According to the reviewer’s suggestion,the hydrophobicity (%) was replaced to “Cell surface hydrophobicity (%)”.

Chapter 3.2.1.2

  1. Please, complete the sentence “The co-aggregation of the isolates is (Figure 1B)”

Authors’ response: According to the reviewer’s suggestion, the sentence was completed by adding “as shown in figure 1B”.

Figure 1 B

  1. On X-axis are not "isolates"!!! It is "% of co-aggregation". The isolates are on Y-axis!

Please enlarge the graph, e.g. by putting the legend below it. Maybe you should use the same way of data presentation on Figure 1A and 1B? With the same data on X and Y-axis.

Authors’ response: According to the reviewer’s suggestion, the graph was re-plotted and replaced with the new figure.

Chapter 3.2.1.3

  1. “…all of the strains had a similar ability as the chicken epithelial cell. (Figure 2) The…”

Authors’ response: According to the reviewer’s suggestion, the sentence was deleted to avoid further confusions.

  1. HT-29 line is a human cell line!

Authors’ response: According to the reviewer’s query, the HT-29 is human colorectal adenocarcinoma cell line.

Figure 2.

  1. Improve the figure description.

I suggest: LAB strains adhesion to buccal epithelial cells observed under a light microscope. (A) Buccal epithelial cells (Control). The adhesion of isolate: RAMULAB07 (B), RAMULAB08 (C) …. to buccal epithelial cells.

What the black arrows indicate?

Authors’ response: According to the reviewer’s suggestion, the figure description was improved and the figure was replaced the indication of black arrow is given in the note.

Table 4

  1. Are differences statistically significant?

Authors’ response: The statistical significance has been removed and the table has been expressed in Mean ± SD values.

Figure 3

  1. Lack of statistical analysis on graphs! Are the differences significant?

Authors’ response: The authors would like to mention that the results have not been compared with the control.

Chapter 3.2.2.2

  1. “All the six isolates were capable of optimum growth. Later, the growth gradually decreased as the time of incubation increased…. The isolate had the most favourable viability rate, and survival rate of above 98% up to a period of 8 h….” – I don’t’ understand.

Authors’ response: According to the reviewer’s query, the sentence was rephrased to avoid confusions

Figure 4.

  1. Correct: Survival rate of isolates in gastric (A) and intestinal (B) juice.

Authors’ response: According to the reviewer’s suggestion, the figure description was corrected.

Chapter 3.3.1

  1. Correct into: The isolates were tested for antimicrobial activity against the pathogenic bacteria. … The zone of inhibition scale ranges from 6- 20 mm.

Authors’ response: According to the reviewer’s suggestion, the sentence was corrected in the manuscript.

Chapter 3.3.2

  1. The isolates were tested against 13 antibiotics to determine the antibiotic resistance profile. → To determine the antibiotic resistance profile, the isolates were tested against 13 antibiotics.

Authors’ response: According to the reviewer’s suggestion, the sentence was replaced in the manuscript.

Table 6

  1. Add reference “CLSI, 2018” to the bibliography list.

Authors’ response: According to the reviewer’s suggestion, the referencewas added in the manuscript.

Chapter 3.3.3.

  1. The sentences: “…the six LAB isolates showed no zone around the colonies and therefore considered as safe classified as γ-hemolysis. The appearance of a clear zone indicates that the isolates where safe to utilize. DNAse activity was also a measure to evaluate the safety of the probiotic formulation. The isolated that demonstrated no zone of inhibition confirmed their non-pathogenic character.”

I suggest:

“… the six LAB isolates showed no zone around the colonies and therefore considered as safe and classified as γ-hemolysis. The appearance of a clear zone indicates that the isolates were safe to utilize. DNAse activity was also a measure to evaluate the safety of the probiotic formulation. The isolates that demonstrated no zone of inhibition confirmed their non-pathogenic character.

Authors’ response: According to the reviewer’s suggestion, the sentence wasrephrased.

Chapter 3.4.

  1. The six biochemically characterized isolates LAB were amplified… - change into isolates of LAB or LAB isolates

Authors’ response: According to the reviewer’s suggestion, in the sentence “isolates LAB”was changed to “isolates of LAB”.

Figure 6.

  1. Make these graphs again! The chart type is wrong, as this is not "kinetics". You cannot connect the dots with a line between separate strains. Use the bars like for example in Figure 1A.

Authors’ response: According to the reviewer’s suggestion, the graph design was changed and replaced with the new figure

Chapter 3.6

  1. “In our study, the inhibitory activity against α-glucosidase and α-amylase was per-formed using the CS, CE, and IC of the three isolates.” – Why 3?

Authors’ response: According to the reviewer’s query, the sentence was rephrased to avoid confusions.

  1. “The intact cells of the isolates showed the least inhibition compared to supernatant and pellets“ – explain in discussion WHY?

Authors’ response: According to the reviewer’s query, the intact cells of the isolates showed the least inhibition compared to supernatant and pellets will be discussed in the discussion.

Figure 7.

  1. Alpha glucosidase (A) and alpha amylase (B) inhibitory activity of the isolates. Unify with the text: α-glucosidase, α-amylase…

Authors’response: According to the reviewer’s suggestion, the figure description was changed and the figure was replaced.

Discussion

The majority of Discussion is only the repetition of the obtained results. It should be discussed what such results means for probiotics properties and for human (consumer of probiotics). And compare their properties with other known probiotic strains (maybe commercial). Are RAMULAB strains better or no?

  1. “Many studies have found that phenol tolerance has a high viability rate [24,47,48].” I don’t understand.

Authors’ response: According to the reviewer’s query, the sentence was rephrased to avoid confusions

  1. “Certain digestion processes release hazardous metabolites and like phenol that is produced by….”

Authors’ response: According to the reviewer’s suggestion, the sentence was corrected.

  1. “Because coaggregation is the intercellular adhesion of distinct strains, the probiotic microbe can adhere to the pathogen intracellularly.” How LAB can adhere INTRACELLULARLY to pathogen cell?

Authors ‘response: According to the reviewer’s suggestion, the word intracellularly was replaced with intercellular.

  1. Question to the part of discussion where resistance/susceptibility to antibiotics is described: It is good phenomena that isolates were resistant to some antibiotics, while sensitive to others? What it means for their probiotics properties?

Authors ‘response: According to the reviewer’s query, the isolates are probiotic and they are generally recognized to be safe (GRAS), yet as per 2002 FAO/ WHO guidelines it is recommended to assess the safety of potential probiotic by minimal tests which includes antibiotics resistance patterns.

  1. “… activity of 24.11% for CFS and 10.563 % for CFE were the greatest [26]. ” – explain the abbreviations!

Authors ‘response: According to the reviewer’s suggestion, the abbreviations were added in the sentence.

  1. “In this study the Lactobacillus isolated are from food source has expressed the promising results for the inhibition of α-glucosidase and α-amylase.” Improve English, please.

Authors ‘response: According to the reviewer’s suggestion, the sentence was rephrased.

Conclusion

  1. “In comparison to the IC, the six isolates CS and CE displayed substantial inhibitory activity of α-glucosidase and α-amylase. “ You should explain why (in discussion part)!

Authors ‘response: According to the reviewer’s suggestion, the explanation was added in the discussion part.

Round 2

Reviewer 2 Report

Sorry, but the Authors’ response: "The authors would like to mention that the data obtained in the present study are the mean of triplicate determinations expressed as mean ± SD. Hence, to avoid confusions and to be precise statistical significance has been deleted in the entire manuscript" ... and ..... "and the statistical significance has been removed throughout the manuscript since the results were not compared with the controls to avoid further confusions" is an absurd.

All scientific results require to be verified by statistical analysis! Mean ± SD is only the way of presentation of the results. Only statistical analysis allows the Authors of study state that they had reported differences between samples (of course if they are statistically significant according to the statistical test used. and the selection of the statistical test depends of the kind of results and experiments performed). As at the moment none of the results (tables, reports) contain statistical analysis the obtained results are worthless.

I'm sorry for that, because the authors have done a lot of research and it would be worth to publish it. But not in this form!

Other remarks:

Some conclusions are unjustified. For example "In this study, the CS and CE have a comparatively higher inhibitory effect than the IC..." The Authors cannot infer a greater/lesser effect if it is not known whether the differences were statistically significant.

Some correction are not introduced, despite in the Authors' answer is written that according to the reviewer changes were introduces/made. For example, there is still no data about the enzymes used in the study (producer, units of activity etc.), Cell surface hydrophobicity (%) in Table 3 was not corrected as well as suggested changes in Chapter 3.3.1 were not introduced.

In my opinion the manuscript still requires the "extensive editing of English language", however, I'm not a native. But obvious English errors are still present. Moreover, still a lot of errors are i nthe manuscript: doubled words, lack or too many spaces, lack of italic font when Latin names are used, etc.

Author Response

Response to reviewer comments:

All the authors thank the reviewer for their thoughtful suggestions to improve the quality of the manuscript. The response to all the comments with point-by-point is provided below.

Reviewer 2

Sorry, but the Authors’ response: "The authors would like to mention that the data obtained in the present study are the mean of triplicate determinations expressed as mean ± SD. Hence, to avoid confusions and to be precise statistical significance has been deleted in the entire manuscript" ... and ..... "and the statistical significance has been removed throughout the manuscript since the results were not compared with the controls to avoid further confusions" is an absurd.

All scientific results require to be verified by statistical analysis! Mean ± SD is only the way of presentation of the results. Only statistical analysis allows the Authors of study state that they had reported differences between samples (of course if they are statistically significant according to the statistical test used. And the selection of the statistical test depends of the kind of results and experiments performed). As at the moment none of the results (tables, reports) contain statistical analysis the obtained results are worthless.

I'm sorry for that, because the authors have done a lot of research and it would be worth to publish it. But not in this form!

Response: The authors appreciate the critic comments and suggestions for the statistical analysis. As per the suggestion, the statistical comparisons between the isolates were performed by one-way analysis of variance (ANOVA), followed by Duncan’s Multiple Range Test using SPSS Software. The results were considered statistically significant if the ‘p’ value were 0.05 or less. The graphs were redrawn with proper annotations using the Graph pad Prism and the same was updated in the manuscript.

After the statistical analysis, the authors believe that the manuscript is worth to publish in the journal. Also authors would like to thank the reviewer for his suggestion and gave us a scope to improve the manuscript.

Other remarks:

Some conclusions are unjustified. For example "In this study, the CS and CE have a comparatively higher inhibitory effect than the IC..." The Authors cannot infer a greater/lesser effect if it is not known whether the differences were statistically significant.

Response: As per the reviewer suggestion and to the previous comment, the statistical analysis incorporated to the manuscript has given more weightage. However the authors have remodelled the entire conclusion for better understanding and to be more precise.

The authors would like to mention that “all the six isolates displayed substantial inhibitory activity of α-glucosidase and α-amylase when tested using their IC, CS and CE with IC faring to be the most optimal” and the same has been incorporated in the manuscript which also holds good with the statistical analysis.

Some correction are not introduced, despite in the Authors' answer is written that according to the reviewer changes were introduces/made. For example, there is still no data about the enzymes used in the study (producer, units of activity etc.), Cell surface hydrophobicity (%) in Table 3 was not corrected as well as suggested changes in Chapter 3.3.1 were not introduced.

Response: The authors apologise for the errors made. We have now carefully evaluated the manuscript for any such errors and corrected accordingly.

As per the reviewer’s suggestion, data about the enzymes used in the study (producer, units of activity etc.) have been incorporated in the chapter 2.3.2.2., and the “Cell surface hydrophobicity (%)” in Table 3 has been corrected in the manuscript. All the suggested changes in chapter 3.3.1 has been changed and corrected according to the suggestion. Also,

  • The details of instrument used were updated in the chapter 2.1.
  • In chapter 2.3.3.3 “Using the procedure given by Sorokulova et al. [30] with slight methodology modification, the hemolytic activity of the isolates was tested.” was changed.
  • In chapter 2.3.3.4, the “… reddish zone…”was changed to “zone around the colonies” to avoid further confusions.
  • In chapter 2.5, the statement was rephrased for the better explanations.
  • In chapter 2.6, line 244 the spelling mistake of ‘thr’ was altered to ‘the’.
  • In chapter 2.8, the explanation regarding the statistical analysis has been updated in the manuscript.
  • In chapter 3.3.2, the sentence was corrected from ‘The determine…’ was changed to ‘To determine..’ .
  • The tables, footnote and graphs has been updated throughout the manuscript with the statistically significance according to the statistical test.

In my opinion the manuscript still requires the "extensive editing of English language", however, I'm not a native. But obvious English errors are still present. Moreover, still a lot of errors are in the manuscript: doubled words, lack or too many spaces, lack of italic font when Latin names are used, etc.

Response: As per the reviewer suggestion, the manuscript has been proofread again by a native English speaker to minimize typological and grammatical errors. All the minor errors have been corrected to improve the quality of the manuscript.

All the doubled words, lack or too many spaces, lack of italic font when Latin names are used have been corrected in the entire manuscript to be more precise.

The authors would like to mention that due to the various versions of Microsoft office, the errors have been retained for example the space between the words.